## [Peer Review File · EMBO Reports]

VGLL4 Modulates Paneth Cells and Sustains Intestinal Homeostasis

Haoen Zhang, Zuoyun Wang, Xiaodong Wang, Wentao Yu, Guoying Zhang, Haijiao Zhang, Yi Lu, Yang Sun, Tiantian Lu, Xiaoyu Li, Ruizeng Yang, Jiaqi Sun, Jinjin Xu, Shuo Huang, Xueyan Ma, Jiale Ren, Nan Tang, Zhonghua Cheng, Jing Yu, Fang Wei, Hu Zhou, Jinsong Li, Jun Qin, Yunyun Jin, and Lei Zhang

Corresponding author(s): Lei Zhang (rayzhang@sjtu.edu.cn), Yunyun Jin (yunyunjin@sjtu.edu.cn)

Review Timeline:

Transfer Date:	30th Jul 25
Editorial Decision:	19th Aug 25
Revision Received:	17th Nov 25
Editorial Decision:	8th Dec 25
Revision Received:	19th Dec 25
Accepted:	15th Jan 26

Editor: Achim Breiling

Transaction Report: This manuscript was transferred to EMBO reports following peer review at The EMBO Journal.

Referee #1:

Vestigial-like family member 4 (VGLL4) is a transcriptional coactivator known to interact with TEAD family of proteins to regulate Hippo pathway. This paper first demonstrated that VGLL4 is reduced in intestinal epithelia upon irradiation and DSS-induced injuries in mice, and in human ulcerative colitis samples. The authors then characterized the intestinal phenotypes of mice lacking epithelial VGLL4. These mice have reduced Paneth cells and defensin gene expression, gut microbial alteration, and reduced regenerative capacity in vivo and in organoid culture. The authors performed co-IP assays to show an interaction of VGLL4 with TEAD4 and ATOH1, and a chromatin occupancy of TEAD4 and ATOH1 at GFI1 locus. Using a luciferase-based defensin promoter assay, the authors showed a regulation of defensin by VGLL4, TEAD4 and TCF4. They suggest that VGLL4 regulates Paneth cell defensin, hence affecting microbial diversity. They conclude the identification of a dual role for VGLL4, namely regulating Paneth cell differentiation and for defensin expression.

Overall, while the phenotypic analysis of VGLL4 IEC-knockout mice and the biochemical association of VGLL4-TEAD4-ATOH1 are appreciated, there are some conceptual and experimental limitations that fail to convincingly drive the conclusion.

Major concerns:

1. VGLL4 is a known TEAD coactivator, the complexing of these two is not surprising. In Fig. 4, while I do see from the co-IP experiments displaying a complex of VGLL4 with TEAD4 and ATOH1, there is no evidence to support that VGLL4 is at the same chromatin sites of GFI1 locus where TEAD4 and ATOH1 bind.
2. Although the histochemistry or Western blots are generally well done, the N numbers in several key quantification panels are 3 each group, which is concerning and making the reviewer wonder if these experiments were repeated in a rigorous way.
3. The data used for the claim of a complex formation of VglL4-TEAD4-Tcf4 is not at all convincing. This conclusion seems to be mainly driven by luciferase reporter assays.
4. Without a rescue or restoration experiment in Fig. 6, the microbial data presented in this figure appears to be detached from the main paper. There are numerous alternative reasons for a changed microbiota in these KO mice, which lack a Paneth cell population. The conclusion regarding microbiota is too rough and weak.
5. Fig. 3 and half of Fig. 5 are in fact making the same point about Paneth cells. When there is a diminished Paneth cell population, there certainly won't have Paneth cell enriched genes such as the alpha defensin.
6. Fig. 1J shows an allele of VglL4-3'HA-CreERT2-eGFP, but I do not clearly see a characterization or application of this allele. Panel K shows an HA staining. Is this HA tag in frame with endogenous VglL4 gene? Did the authors validate the CreER function and the

expression of the eGFP cassette in these mice?

Referee #2:

Paneth cells play key roles in intestinal stem cell homeostasis, inflammation and regeneration. But the mechanisms underlying their differentiation are still poorly understood. Zhang et al., performed gene expression analyses and utilized mouse genetics, demonstrating the mechanistic role of the *Vgll4* in Paneth cell differentiation. Their comprehensive biochemical studies provide new insight into the transcriptional regulation of Paneth cell differentiation and defensin expression. The following suggestions might further improve the current manuscript.

Fig 3H shows no detectable Lysozyme protein level in *Vgll4* KO mice, but IF staining shows a ~50% reduction in the number of Paneth cells expressing Lysozyme protein in vivo. Since *Vgll4* expression is restricted to the crypt, the authors should analyze protein levels using isolated crypt cells to address this issue.

In Fig 3I, the authors performed stem cell co-culture with (*Vgll4* WT and KO) Paneth cells and found a defect in organoid growth. This indicates that *Vgll4* KO not only reduces Paneth cell numbers but also impairs their function. From this assay, the authors mentioned that the niche-forming ability of Paneth cells is impaired. Did they use WT stem cells in this co-culture assay? Since *Vgll4* is also expressed in stem cells, it might play a role in stem cells. This should be clarified.

The authors' mechanistic studies in Fig 4 have demonstrated the role of *VGLL4*-*TEAD4*-*ATOH1* in secretory lineage differentiation (including their target *GFI1*). This data is important, providing new insight into secretory lineage differentiation. Interestingly, Yap signaling is essential for intestinal regeneration. The authors discussed the YAP overexpression study, but this may not be useful for addressing its potential role in Paneth cell differentiation. They might have distinct cell type-specific roles, and Yap and *VGLL4* could still cooperate in Paneth cell differentiation. The authors could address this question in organoid culture by using YAP inhibitors (if the authors have no YAP/*Taz* mouse alleles).

The authors mentioned *VGLL4* as a transcription factor in Fig 5. Isn't this a transcriptional cofactor? Does it bind to DNA? This point should be clarified.

In Fig 6, the authors demonstrated that Vgl4 KO induces microbial dysbiosis, but it remains unclear whether it plays a role in intestinal regeneration. Would antibiotic treatment be able to rescue the intestinal regeneration defects?

Referee #1:

Vestigial-like family member 4 (VGLL4) is a transcriptional coactivator known to interact with TEAD family of proteins to regulate Hippo pathway. This paper first demonstrated that VGLL4 is reduced in intestinal epithelia upon irradiation and DSS-induced injuries in mice, and in human ulcerative colitis samples. The authors then characterized the intestinal phenotypes of mice lacking epithelial VGLL4. These mice have reduced Paneth cells and defensin gene expression, gut microbial alteration, and reduced regenerative capacity in vivo and in organoid culture. The authors performed co-IP assays to show an interaction of VGLL4 with TEAD4 and ATOH1, and a chromatin occupancy of TEAD4 and ATOH1 at *GFI1* locus. Using a luciferase-based defensin promoter assay, the authors showed a regulation of defensin by VGLL4, TEAD4 and TCF4. They suggest that VGLL4 regulates Paneth cell defensin, hence affecting microbial diversity. They conclude the identification of a dual role for VGLL4, namely regulating Paneth cell differentiation and for defensin expression.

Thank you for your thoughtful assessment of our manuscript. In response to your feedback, we have now incorporated additional data to further substantiate our conclusions.

Overall, while the phenotypic analysis of VGLL4 IEC-knockout mice and the biochemical association of VGLL4-TEAD4-ATOH1 are appreciated, there are some conceptual and experimental limitations that fail to convincingly drive the conclusion.

Major concerns:

1. VGLL4 is a known TEAD coactivator, the complexing of these two is not surprising. In Fig. 4, while I do see from the co-IP experiments displaying a complex of VGLL4 with TEAD4 and ATOH1, there is no evidence to support that VGLL4 is at the same chromatin sites of *GFI1* locus where TEAD4 and ATOH1 bind.

Thank you for your insightful comment. To further demonstrate that VGLL4 binds to the same chromatin site and regulates the *GFI1* transcriptional level, we overexpressed SBP-tagged VGLL4 in 293T cells and conducted a ChIP assay using SBP-VGLL4. Results showed that SBP-VGLL4 bound to the same site as we used the same ChIP primers at *GFI1* promoter (Figure 5H). Furthermore, we proved that by overexpressing VGLL4-TEAD4-ATOH1 complex, the binding of TEAD4, ATOH1 and VGLL4 to the *GFI1* promoter site was all upregulated (Figure 5F-H). Overexpression of VGLL4 alone also upregulated the binding of endogenous TEAD4 and ATOH1 to the *GFI1* site

(Figure S5K-S5L). These data indicate that the VGLL4-TEAD4-ATOH1 complex binds to the promoter of *GF11* to activate its transcription.

Figure 5

Figure S5

(Figure 5F) ChIP-qPCR analysis of HA-TEAD4 on *GF11* promoter in 293T cells with or without VGLL4 and ATOH1 overexpression. Three biological replicates per group. (Figure 5G) ChIP-qPCR analysis of MYC-ATOH1 on *GF11* promoter in 293T cells with or without VGLL4 and TEAD4 overexpression. Three biological replicates per group. (Figure 5H) ChIP-qPCR analysis of SBP-VGLL4 on *GF11* promoter in 293T cells with or without ATOH1 and TEAD4 overexpression. Three biological replicates per group. (Figure S5K) ChIP-qPCR analysis of TEAD4 on *GF11* promoter in 293T cells with or without VGLL4 overexpression. Three biological replicates per group. (Figure S5L) ChIP-qPCR analysis of ATOH1 on *GF11* promoter in 293T cells with or without VGLL4 overexpression. Three biological replicates per group.

2. Although the histochemistry or Western blots are generally well done, the N numbers in several key quantification panels are 3 each group, which is concerning and making the reviewer wonder if these experiments were repeated in a rigorous way.

Thank you for highlighting this crucial point. We completely agree that, and we have increased the sample size (N) of mice in Figure 2C, 3B, 3D, 3F, 3I, 6A and 6D.

Figure 2

Figure 3

Figure 3

Figure 6

(Figure 2C-D) Representative images (C) and quantification (D) of small intestinal organoids from *Vgll4^{fl/fl}* and *Vgll4^{IEC-KO}* mice (n=5). Scale bars, 50 μ m.

(Figure 3B) IF staining of Lysozyme (green) in *Vgll4^{fl/fl}* and *Vgll4^{IEC-KO}* mice. EpCAM, gray. DAPI, blue. Scale bars, 20 μ m. (Figure 3C) Quantification of Lysozyme⁺ Paneth cells in *Vgll4^{fl/fl}* and *Vgll4^{IEC-KO}* mice crypts (n=6). (Figure 3D) IF staining of Defa5 (green) in *Vgll4^{fl/fl}* and *Vgll4^{IEC-KO}* mice. EpCAM, gray. DAPI, blue. Scale bars, 20 μ m. (Figure 3E) Quantification of Defa5⁺ Paneth cells in *Vgll4^{fl/fl}* and *Vgll4^{IEC-KO}* mice crypts (n=6). (Figure 3F) Flow cytometry analysis of Paneth cells from *Vgll4^{fl/fl}* and *Vgll4^{IEC-KO}* mice. (Figure 3G) Quantification of Paneth cell proportion in *Vgll4^{fl/fl}* and *Vgll4^{IEC-KO}* mice (n=6). (Figure 3I) Representative pictures of organoids from wild-type Lgr5⁺ stem cells co-cultured with Paneth cells from *Vgll4^{fl/fl}* and *Vgll4^{IEC-KO}* mice for 5 days. Scale bars, 60 μ m. (Figure 3J) Quantification of organoids number in (I) (n=6). (Figure 3K) Quantification of organoids diameter in (I) (n=6). (Figure 6A) Electron microscopy images of granules in Paneth cells in the ileum crypts in *Vgll4^{fl/fl}* and *Vgll4^{IEC-KO}* mice. Scale bar, 1 μ m. (Figure 6B) Quantification of granule diameters in (A) (n=5). (Figure 6D) qRT-PCR analysis of Lysozyme and defensin mRNA levels in ileum samples from *Vgll4^{fl/fl}* and *Vgll4^{IEC-KO}* mice (n=6).

3. The data used for the claim of a complex formation of Vgll4-TEAD4-Tcf4 is not at all convincing. This conclusion seems to be mainly driven by luciferase reporter assays.

We sincerely appreciate your constructive suggestions. We used co-IP to confirm the formation of the VGLL4-TEAD4-TCF4 complex (Figure 6F). Using ChIP, we detected the binding of TCF4, TEAD4, and VGLL4 on *DEFA5* promoter (Figure 6G-I, Figure S6B-S6C). Overexpression of the TCF4-TEAD4-VGLL4 complex upregulated the binding of TCF4 to the *DEFA5* promoter (Figure 6J). These data further strengthen the notion that the VGLL4-TEAD4-TCF4 complex regulates the expression of defensins.

Figure 6

Figure S6

(Figure 6F) Co-immunoprecipitation assay results between TCF4, TEAD4 and VGLL4 in HEK293T cells. (Figure 6G) ChIP-qPCR analysis of TCF4 enrichment at the *DEFA5* promoter in HCoEpiC cells. Three biological replicates per group. (Figure 6H) ChIP-qPCR analysis of TEAD4 enrichment at the *DEFA5* promoter in HCoEpiC cells. Three biological replicates per group. (Figure 6I) ChIP-qPCR analysis of VGLL4 enrichment at the *DEFA5* promoter in HCoEpiC cells. Three biological replicates per group. (Figure 6J) ChIP-qPCR analysis of FLAG-TCF4 on *DEFA5* promoter in 293T cells with or without VGLL4 and TEAD4 overexpression. Three biological replicates per group. (Figure S6B) ChIP-qPCR analysis of TCF4 enrichment at the *DEFA5* promoter in SW620 cells. Three biological replicates per group. (Figure S6C) ChIP-qPCR analysis of TEAD4 enrichment at the *DEFA5* promoter in SW620 cells. Three biological replicates per group.

4. Without a rescue or restoration experiment in Fig. 6, the microbial data presented in this figure appears to be detached from the main paper. There are

numerous alternative reasons for a changed microbiota in these KO mice, which lack a Paneth cell population. The conclusion regarding microbiota is too rough and weak.

We appreciate your insightful comments. To completely clear the microbiome, we used antibiotics (working concentration: ampicillin 1g/L, metronidazole 1g/L, vancomycin hydrochloride 500mg/L, neomycin 1g/L in drinking water) to treat *Vgll4^{fl/fl}* and *Vgll4^{IEC-KO}* mice before DSS treatment for 4 weeks, then added 2%DSS in the drinking water for 5 days, then harvested the colon. Results showed *Vgll4^{IEC-KO}* mice still exhibited greater body weight loss, shorter colon length, and more severe damage than *Vgll4^{fl/fl}* mice (Response Figure 5A-5D). Considering these results, we believe that these phenotypes are primarily due to the loss of VGLL4 in epithelial cell, instead of microbiome changes. The dysbiosis in the microbiome appeared to be the results of changes in defensins and other antimicrobial peptides, not the reason of disrupted intestinal homeostasis. Now we rearranged the figures and moved the analysis of microbiota to Supplemental figure 7, where it is used solely to describe the changes in microbiome alteration after VGLL4 loss.

Response Figure 5

(Response Figure 5A) Body weight change of *Vgll4^{fl/fl}* and *Vgll4^{IEC-KO}* mice during DSS injury after Abx treatment (n=4). (Response Figure 5B) Representative images of colon samples from *Vgll4^{fl/fl}* and *Vgll4^{IEC-KO}* mice after Abx and DSS treatment. Scale bar, 1 cm. (Response Figure 5C) Quantification

of colon length after Abx and DSS treatment in *Vgll4^{fl/fl}* and *Vgll4^{IEC-KO}* mice. (n=4). (Response Figure 5D) H&E staining of the colons from *Vgll4^{fl/fl}* and *Vgll4^{IEC-KO}* mice after Abx and DSS treatment. Scale bars, 10 μ m.

5. Fig. 3 and half of Fig. 5 are in fact making the same point about Paneth cells. When there is a diminished Paneth cell population, there certainly won't have Paneth cell enriched genes such as the alpha defensin.

We appreciate your insightful comments. In Figure 6, we focus on individual Paneth cells. The decrease in defensins was not just because the decreased number of Paneth cells, but also due to the decreased expression in single Paneth cells. In Figure 6E, we sorted out pure Paneth cells from *Vgll4^{fl/fl}* and *Vgll4^{IEC-KO}* mice. Using qRT-PCR, we normalized the effect of Paneth cell number. Results showed that, even with the same number of Paneth cells, the expression of defensins and Lysozyme was still decreased, indicating that loss of VGLL4 decreased the antimicrobial peptides expression in individual Paneth cells.

Figure 6

(Figure 6E) qRT-PCR analysis of *Lysozyme*, *Mmp7*, *GF11*, and *VGLL4* from sorted Paneth cells from *Vgll4^{fl/fl}* and *Vgll4^{IEC-KO}* mice. Three biological replicates per group.

6. Fig. 1J shows an allele of *Vgll4*-3'HA-CreERT2-eGFP, but I do not clearly see a characterization or application of this allele. Panel K shows an HA staining. Is this HA tag in frame with endogenous *Vgll4* gene? Did the authors validate the CreER function and the expression of the eGFP cassette in these mice?

Thank you for your insightful comment. In Figure 1K we used both an endogenous VGLL4 antibody and a HA antibody to verify the same localization of VGLL4 and VGLL4-HA. Then in Figure S2D-S2G, we used two different mouse lines: one was *Vgll4*-3*HA-CreERT2-eGFP, *A19^{+/-}* mice, TAM injection for 4 weeks, resulting in tdTomato signal in both the whole intestine and organoids. Another one was *Vgll4*-3*HA-CreERT2-eGFP, *mTmG^{+/-}* mice. After TAM injection for 7 days, Cre-expressing cells exhibited GFP signal, while cells where the Cre didn't work still retained the tdTomato signal. IF of both mouse

small intestines showed remarkable efficiency of CreERT2. The endogenous eGFP cassette intensity was not strong enough for observation (Response Figure 1A), so we used an anti-GFP antibody to magnify the signal (Response Figure 1B). Inside our research we didn't use this endogenous GFP as any marker.

Figure 1K

Figure S2

Response Figure 1

(Figure 1K) Dual-IF for VGLL4 (green) and HA (gray) in crypts of *Vgll4-3*HA-CreERT2-eGFP* mice. Scale bars, 20 μ m. (Figure S2D) IF of VGLL4 (gray) and tdTomato (red) in small intestinal organoids from *Vgll4-3*HA-CreERT2-eGFP*, *A19^{+/-}* mice injected with TAM for 4 weeks. Scale bars, 40 μ m. (Figure S2E) IF of tdTomato (red) in *Vgll4-3*HA-CreERT2-eGFP*, *A19^{+/-}* mice injected with TAM for 4 weeks. Scale bars, 30 μ m. (Figure S2F) Model of *Vgll4-3*HA-CreERT2-eGFP*, *mTmG^{+/-}* mouse construction strategy and small intestinal samples harvest strategy. (Figure S2G) IF of VGLL4 (gray), tdTomato (purple) and GFP (green) in *Vgll4-3*HA-CreERT2-eGFP*, *mTmG^{+/-}* mouse small intestinal samples after TAM injection for a week. Scale bars, 25 μ m. (Response Figure 1A) Endogenous GFP signal (green) in *Vgll4-3*HA-CreERT2-eGFP* mice. DAPI,

blue. Scale bars, 30 μ m. (Response Figure 1B) IF of VGLL4-HA (gray) and GFP (green) in *Vgll4-3*HA-CreERT2-eGFP* mice. DAPI, blue. Scale bars, 15 μ m.

Referee #2:

Paneth cells play key roles in intestinal stem cell homeostasis, inflammation and regeneration. But the mechanisms underlying their differentiation are still poorly understood. Zhang et al., performed gene expression analyses and utilized mouse genetics, demonstrating the mechanistic role of the *Vgll4* in Paneth cell differentiation. Their comprehensive biochemical studies provide new insight into the transcriptional regulation of Paneth cell differentiation and defensin expression. The following suggestions might further improve the current manuscript.

Thank you for your assessment of our manuscript. In response to your feedback, we have now incorporated new data to further substantiate our conclusions.

Fig 3H shows no detectable Lysozyme protein level in *Vgll4* KO mice, but IF staining shows a ~50% reduction in the number of Paneth cells expressing Lysozyme protein in vivo. Since *Vgll4* expression is restricted to the crypt, the authors should analyze protein levels using isolated crypt cells to address this issue.

We sincerely appreciate your constructive suggestions. As suggested, we used the isolated crypts to do the WB. Our two independent results (Response Figure 2A-2B) showed consistent trends in the protein levels of Lysozyme1, GF11, SOX9, and ATOH1, similar to those observed in ileum samples. Figure 3H was from the Response Figure 2A.

Response Figure 2

Figure 3H

(Response Figure 2A) Immunoblot analysis of indicated proteins in small intestinal crypt samples of *Vgll4^{fl/fl}* and *Vgll4^{IEC-KO}* mice (n=3). (Response Figure 2B) Immunoblot analysis of indicated proteins in small intestinal crypt samples of *Vgll4^{fl/fl}* and *Vgll4^{IEC-KO}* mice (n=3). (Figure 3H) Immunoblot analysis of VGLL4 and Lysozyme in small intestinal crypt samples of *Vgll4^{fl/fl}* and *Vgll4^{IEC-KO}* mice (n=3).

In Fig 3I, the authors performed stem cell co-culture with (*Vgll4* WT and KO) Paneth cells and found a defect in organoid growth. This indicates that *Vgll4* KO not only reduces Paneth cell numbers but also impairs their function. From

this assay, the authors mentioned that the niche-forming ability of Paneth cells is impaired. Did they use WT stem cells in this co-culture assay? Since *Vgll4* is also expressed in stem cells, it might play a role in stem cells. This should be clarified.

We sincerely appreciate your constructive suggestions. In this co-culture assay, wild-type *Lgr5*⁺ stem cells (WT *Lgr5*⁺) were sorted from *Lgr5-creERT2-eGFP*^{+/-} mice, and VGLL4 knockout *Lgr5*⁺ stem cells (VGLL4-KO *Lgr5*⁺) were sorted from *Lgr5-creERT2-eGFP*^{+/-}, *Vgll4*^{fl/fl} mice. All *Lgr5-creERT2-eGFP*^{+/-} mice and *Lgr5-creERT2-eGFP*^{+/-}, *Vgll4*^{fl/fl} mice were injected with TAM three times within one week. Wild-type Paneth cells (WT PCs) were sorted from *Vgll4*^{fl/fl} mice, and VGLL4 knockout Paneth cells (VGLL4-KO PCs) were sorted from *Vgll4*^{IEC-KO} mice. We set four groups: WT *Lgr5*⁺ with WT PCs, WT *Lgr5*⁺ with VGLL4-KO PCs, VGLL4-KO *Lgr5*⁺ with WT PCs, VGLL4-KO *Lgr5*⁺ with VGLL4-KO PCs. WT *Lgr5*⁺ cells co-cultured with WT PCs resulted in the largest number and diameter of organoids, while VGLL4-KO *Lgr5*⁺ cells co-cultured with VGLL4-KO PCs formed the fewest and smallest organoids. Compared to the organoids formed by WT *Lgr5*⁺ cells co-cultured with WT PCs, VGLL4-KO *Lgr5*⁺ cells co-cultured with WT PCs formed fewer organoids, and showed a tendency towards smaller diameters (Response Figure 3A-3C). This indicates that VGLL4 also plays a role in stem cells. Co-culture of WT *Lgr5*⁺ cells with VGLL4-KO PCs formed smaller organoids than those formed by VGLL4-KO *Lgr5*⁺ cells with WT PCs, indicating that the knockout of VGLL4 in PCs had a greater influence on organoids formation than in *Lgr5*⁺ stem cells. Figure 3I-3K showed the results of WT *Lgr5*⁺ cells co-cultured with either WT PCs or VGLL4-KO PCs in Response Figure3.

Response Figure 3

Figure 3

(Response Figure 3A) Representative pictures of organoids from wild type Lgr5⁺ stem cells (WT Lgr5⁺) or VGLL4 Knockout Lgr5⁺ stem cells (VGLL4-KO Lgr5⁺) co-cultured with Paneth cells from *Vgll4^{fl/fl}* and *Vgll4^{IEC-KO}* mice for 5 days. Scale bars, 60 μm. (Response Figure 3B) Quantification of organoids number in (Response Figure 3A) (n=6). (Response Figure 3C) Quantification of organoids diameter in (Response Figure 3A) (n=6). (Figure 3I) Representative pictures of organoids from wild-type Lgr5⁺ stem cells co-cultured with Paneth cells from *Vgll4^{fl/fl}* and *Vgll4^{IEC-KO}* mice for 5 days. Scale bars, 60 μm. (Figure 3J) Quantification of organoids number in (Figure 3I) (n=6). (Figure 3K) Quantification of organoids diameter in (Figure 3I) (n=6).

The authors' mechanistic studies in Fig 4 have demonstrated the role of VGLL4-TEAD4-ATO1 in secretory lineage differentiation (including their target GF11). This data is important, providing new insight into secretory lineage differentiation. Interestingly, Yap signaling is essential for intestinal regeneration. The authors discussed the YAP overexpression study, but this may not be useful for addressing its potential role in Paneth cell differentiation. They might have distinct cell type-specific roles, and Yap and VGLL4 could still cooperate in Paneth cell differentiation. The authors could address this question in organoid culture by using YAP inhibitors (if the authors have no YAP/Taz mouse alleles).

We sincerely appreciate your constructive suggestions. As suggested, we used

YAP inhibitor Verteporfin (VP) to culture the organoids from *Vgll4^{fl/fl}* and *Vgll4^{IEC-KO}* mice. Given that previously YAP was reported to play an important role in stem cell proliferation, our results also showed that the addition of VP impaired organoid formation, crypts treated with VP were unable to form organoids. In our model, YAP plays a critical role upstream in maintaining and promoting stem cell proliferation, while VGLL4 functions as a downstream regulator of the differentiation process.

Response Figure 4

(Response Figure4) Representative pictures of organoids from *Vgll4^{fl/fl}* and *Vgll4^{IEC-KO}* mice cultured with DMSO or Verteporfin (VP, 3µM). Scale bars, 200 µm.

The authors mentioned VGLL4 as a transcription factor in Fig 5. Isn't this a transcriptional cofactor? Does it bind to DNA? This point should be clarified.

We apologize for this mistake, as VGLL4 is a transcriptional cofactor. We have updated the manuscript and corrected all relevant sentences.

In Fig 6, the authors demonstrated that *Vgll4* KO induces microbial dysbiosis, but it remains unclear whether it plays a role in intestinal regeneration. Would antibiotic treatment be able to rescue the intestinal regeneration defects?

We appreciate your insightful comments. As suggested, we used antibiotics (working concentration: ampicillin 1g/L, metronidazole 1g/L, vancomycin hydrochloride 500mg/L, neomycin 1g/L in drinking water) to treat *Vgll4^{fl/fl}* and *Vgll4^{IEC-KO}* mice for 4 weeks before DSS treatment to clear the microbiome. We then added 2% DSS to the drinking water for 5 days and harvested the colon. Results showed *Vgll4^{IEC-KO}* mice still exhibited greater body weight loss, shorter colon length, and more severe damage than *Vgll4^{fl/fl}* mice (Response Figure 5A-5D). Based on these results, we believe that these phenotypes were primarily due to the loss of VGLL4 in epithelial cells, rather than changes in the microbiome. The dysbiosis observed in microbiome could be the results of changes in defensins and other antimicrobial peptides, not the reason of

disrupted intestinal homeostasis. We have now rearranged the figures and moved the analysis of microbiota to Supplemental figure 7, where it is used to describe the changes in microbiome alteration after VGLL4 loss.

Response Figure 5

(Response Figure 5A) Body weight change of *Vgll4^{fl/fl}* and *Vgll4^{IEC-KO}* mice during DSS injury after Abx treatment (n=4). (Response Figure 5B) Representative images of colon samples from *Vgll4^{fl/fl}* and *Vgll4^{IEC-KO}* mice after Abx and DSS treatment. Scale bar, 1 cm. (Response Figure 5C) Quantification of colon length after Abx and DSS treatment in *Vgll4^{fl/fl}* and *Vgll4^{IEC-KO}* mice. (n=4). (Response Figure 5D) H&E staining of the colons from *Vgll4^{fl/fl}* and *Vgll4^{IEC-KO}* mice after Abx and DSS treatment. Scale bars, 10 μ m.

Dear Prof. Zhang,

Thank you for the transfer of your revised manuscript from The EMBO Journal to EMBO reports. I have already forwarded to you the reports from the 2 referees that I asked to re-evaluate your study, you will find again below.

I also have received your provisional point-by-point-response (further revision plan). After looking through this, I decided to invite a final revised manuscript that addresses the remaining referee points as indicated in your revision plan. Please also provide a detailed final point-by-point-response to the remaining referee points (see below).

Acceptance of your manuscript will depend on the positive outcome of a final round of review involving referee #2. Revised manuscripts should be submitted within three months of a request for revision. Please contact me to discuss the revision (also by video chat) if you have questions or comments regarding the revision, or should you need additional time.

The manuscript now needs also formatting according to our journal style. Therefore, please carefully review the instructions that follow below.

- 1) a .docx formatted version of the final manuscript text (including legends for main figures, EV figures and tables), but without the figures included. Figure legends should be compiled at the end of the manuscript text.
- 2) individual production quality figure files as .eps, .tif, .jpg (one file per figure), of main figures and EV figures. Please upload these as separate, individual files upon re-submission.

The Expanded View format, which will be displayed in the main HTML of the paper in a collapsible format, has replaced the Supplementary information. You can submit up to 6 images as Expanded View. Please follow the nomenclature Figure EV1, Figure EV2 etc. The figure legend for these should be included in the main manuscript document file in a section called Expanded View Figure Legends after the main Figure Legends section. Additional Supplementary material should be supplied as a single pdf file labeled Appendix. The Appendix should have page numbers and needs to include a table of content on the first page (with page numbers) and legends for all content. Please follow the nomenclature Appendix Figure Sx, Appendix Table Sx etc. throughout the text, and also label the figures and tables according to this nomenclature.

- 3) a .docx formatted letter INCLUDING the reviewers' reports and your detailed point-by-point responses to the remaining concerns. As part of the EMBO Press transparent editorial process, the point-by-point response is part of the Review Process File (RPF), which will be published alongside your paper.

- 4) a complete author checklist, which you can download from our author guidelines

(<https://www.embopress.org/page/journal/14693178/authorguide>). Please insert page numbers in the checklist to indicate where the requested information can be found in the manuscript. The completed author checklist will also be part of the RPF.

- 5) that primary datasets produced in this study (e.g. RNA-seq, ChIP-seq, structural and array data) are deposited in an appropriate public database. If no primary datasets have been deposited, please also state this in a dedicated section (e.g. 'No primary datasets have been generated and deposited'), see below.

The accession numbers and database should be listed in a formal "Data Availability" section that follows the model below. This is now mandatory (like the COI statement). Please note that the Data Availability Section is restricted to new primary data that are part of this study. This section is mandatory. As indicated above, if no primary datasets have been deposited, please state this in this section

Data availability

6) We now request the publication of original source data with the aim of making primary data more accessible and transparent to the reader. You will receive a separate email with instructions for providing source data with your revised manuscript, including information how to upload and organize the files.

8) Regarding data quantification and statistics, please make sure that the number "n" for how many independent experiments were performed, their nature (biological versus technical replicates), the bars and error bars (e.g. SEM, SD) and the test used to calculate p-values is indicated in the respective figure legends (also for EV and Appendix figures). Please also check that all the p-values are explained in the legend, and that these fit to those shown in the figure. Please provide statistical testing where applicable. Please avoid the phrase 'independent experiment', but clearly state if these were biological or technical replicates. Please also indicate (e.g. with n.s.) if testing was performed, but the differences are not significant. In case n=2, please show the data as separate datapoints without error bars and statistics. See also: <http://www.embopress.org/page/journal/14693178/authorguide#statisticalanalysis>

9) Please add scale bars of similar style and thickness to all microscopic images, using clearly visible black or white bars (depending on the background). Please place these in the lower right corner of the images themselves. Please do not write on or near the bars in the image but define the size in the respective figure legend.

10) Please also note our reference format:

12) We now use CRediT to specify the contributions of each author in the journal submission system. CRediT replaces the author contribution section. Please use the free text box to provide more detailed descriptions and do NOT provide your final manuscript text file with an author contributions section. See also our guide to authors: <https://www.embopress.org/page/journal/14693178/authorguide#authorshipguidelines>

13) All Materials and Methods need to be described in the main text using our 'Structured Methods' format, which is required for all research articles. According to this format, the Methods section should include a Reagents and Tools Table (listing key reagents, experimental models, software, and relevant equipment and including their sources and relevant identifiers), uploaded as separate file, and a Methods section in which we encourage the authors to describe their methods using a step-by-step protocol format with bullet points, to facilitate the adoption of the methodologies across labs. More information on how to adhere

to this format as well as downloadable templates (.doc) for the Reagents and Tools Table can be found in our author guidelines (section 'Structured Methods'):

14) Please order the manuscript sections like this, using only these names:

Title page - Abstract (in present tense) - Keywords - Introduction - Results - Discussion - Methods - Data availability section - Acknowledgements (please put here all the funding information) - Disclosure and Competing Interests Statement - References - Figure legends - Expanded View Figure legends

Please also provide a final title with not more than 100 characters including spaces.

15) Please make sure that all the funding information is also entered into the online submission system and that it is complete and similar to the one in the acknowledgement section of the manuscript text file.

16) Please confirm that for all Western blot panels in the manuscript the loading control was run on the same gel as the other proteins detected. Please note that we discourage comparisons between samples on different gels/blots, even if the samples derive from one experiment, as confounding factors reduce comparability. If unavoidable, the figure legend must state that the samples derive from the same experiment and that gels/blots were processed in parallel. If a 'representative' loading control is shown for multiple gels/blots, the intra-gel controls should be shown in the source data files and the figure legends should describe the data displayed accurately. See our author guidelines:

<https://www.embopress.org/page/journal/14693178/authorguide#datapresentationformat> (section 'Electrophoretic gels and blots').

and

<https://www.embopress.org/image-integrity>

Moreover, please note that corresponding authors are required to supply an ORCID ID upon submission of a revised manuscript. Please do this for co-corresponding author Yunyun Jin. Please find instructions on how to link the ORCID ID to the account in our manuscript tracking system in our Author guidelines:

<http://www.embopress.org/page/journal/14693178/authorguide#authorshipguidelines>

In addition, I would need from you uploaded separately:

I look forward to seeing the further revised version of your manuscript when it is ready.

Yours sincerely,

Referee #1:

Since the authors have addressed all of my concerns, I recommend this manuscript for publication.

Referee #2:

The study reported that the expression level of VGLL4 in small intestines changes after irradiation or 2% DSS treatment. Mice with intestinal epithelial Vgll4 deletion is more susceptible to irradiation or DSS induced injury. There is a reduced Paneth cell population in the intestinal Vgll4 knockout mice. VGLL4, TEAD4 and ATOH1 form complex in overexpression system, and regulate GF11 expression in luciferase reporter assays. The last main figure shows that VGLL4, TEAD4 and TCF4 form a complex when overexpressed, and may regulate defensin genes, which may contribute to the changed microbiota shown in supplementary figure.

There are several major concerns:

1. In abstract, authors stated that "Consistently, a remarkable reduction in VGLL4 expression was observed in human ulcerative colitis specimens." The human VGLL4 staining data were presented in Fig. S1B-C. The Method and Result sections indicate that "small intestinal tissues from patients were collected from Department of Gastroenterology, Shanghai Xuhui Central Hospital", and these were "healthy controls (n=5) and acute inflammatory patient samples of small intestines (n=9)". Ulcerative colitis are chronic inflammatory diseases in the colon, the analysis of acute inflammatory small intestines presents a contradiction to the statement in abstract.
2. The manuscript tried to link the disease susceptibility of Vgll4 intestinal knockout mice to Paneth cell defect, as suggested by the title. But the experiments could not establish a causality. Authors used the DSS model that is well known for exclusively inducing inflammation in the distal colon of the mice. The impact on the small intestines in this model likely reflects a secondary effect caused by the systemic illness of the mice. The authors did not provide explanation why they analyzed ileum of DSS-treated mice on day 0, 8, and 11 while the primary disease occurs in colon. It is difficult to convince the readers that the reduced VGLL4 in ileum of these mice worsened the disease in the colon. The Villin-Cre used in their study also delete Vgll4 in the colon. Fig. 2 combines two injury models. Neither was characterized in a sufficient depth.
3. From Fig. 1, VGLL4 is expressed in both Paneth cells and non-Paneth crypt cells, including the stem cells. Therefore, the observed phenotypes in both injury models may be directly caused by a stem cell defect rather than Paneth cells. However, the stem cells in these mice are not well characterized.
4. The biochemistry in Fig. 4 was well done. However, the significance of these results is not very obvious.

Referee #1:

Since the authors have addressed all of my concerns, I recommend this manuscript for publication.

Thank you for your positive feedback and recommendation. We sincerely appreciate your time and valuable comments, which have greatly improved our manuscript. We are grateful for your support.

Referee #2:

The study reported that the expression level of VGLL4 in small intestines changes after irradiation or 2% DSS treatment. Mice with intestinal epithelial Vgll4 deletion is more susceptible to irradiation or DSS induced injury. There is a reduced Paneth cell population in the intestinal Vgll4 knockout mice. VGLL4, TEAD4 and ATOH1 form complex in overexpression system, and regulate GFI1 expression in luciferase reporter assays. The last main figure shows that VGLL4, TEAD4 and TCF4 form a complex when overexpressed, and may regulate defensin genes, which may contribute to the changed microbiota shown in supplementary figure.

Thank you for your thoughtful assessment of our manuscript. In response to your feedback, we have now incorporated additional data to further substantiate our conclusions.

There are several major concerns:

1. In abstract, authors stated that "Consistently, a remarkable reduction in VGLL4 expression was observed in human ulcerative colitis specimens." The human VGLL4 staining data were presented in Fig. S1B-C. The Method and Result sections indicate that "small intestinal tissues from patients were collected from Department of Gastroenterology, Shanghai Xuhui Central Hospital", and these were "healthy controls (n=5) and acute inflammatory patient samples of small intestines (n=9).

Ulcerative colitis are chronic inflammatory diseases in the colon, the analysis of acute inflammatory small intestines presents a contradiction to the statement in abstract.

We sincerely appreciate your thorough evaluation and apologize for any misunderstanding that may have arisen from our original presentation. Our human sample data consist of two parts: small intestinal tissue samples collected from hospital and data analysis from the GSE database. We adjusted our description in abstract to "Analysis of public GSE datasets consistently revealed a significant downregulation of VGLL4 expression in human ulcerative

colitis specimens” to avoid the unclearness. In Figure EV1B-C we detected a decrease of VGLL4 expression in acute inflammatory tissues of the small intestine from patients at the Department of Gastroenterology, Shanghai Xuhui Central Hospital. Additionally, in Fig.EV1D, we analyzed the VGLL4 expression in inflamed, colitis, and UC patient samples from the online GEO database. All the samples showed the decrease of VGLL4 expression, which is consistent with our data from small intestinal tissues.

Figure S1B

Figure S1C

Figure S1D

(Figure EV1B) Immunohistochemistry (IHC) staining of VGLL4 in healthy controls and acute inflammatory patient samples. Scale bars, 20 μ m. (Figure EV1C) Statistical analysis of VGLL4 staining in healthy control (HD) (n=5) and patients (n=9) small intestinal samples. Results are shown as mean \pm SD. (Figure EV1D) GEO analysis of VGLL4 mRNA levels in controls and patient samples. Results are shown as min to max. *p < 0.05, **p < 0.01, ***p < 0.001. Unpaired Student's t-test.

2. The manuscript tried to link the disease susceptibility of Vgll4 intestinal knockout mice to Paneth cell defect, as suggested by the title. But the experiments could not establish a causality. Authors used the DSS model that is well known for exclusively inducing inflammation in the distal colon of the mice. The impact on the small intestines in this model likely reflects a secondary effect caused by the systemic illness of the mice. The authors did not provide explanation why they analyzed ileum of DSS-treated mice on day 0, 8, and 11

while the primary disease occurs in colon. It is difficult to convince the readers that the reduced VGLL4 in ileum of these mice worsened the disease in the colon. The Villin-Cre used in their study also delete Vgll4 in the colon. Fig. 2 combines two injury models. Neither was characterized in a sufficient depth.

We sincerely appreciate the reviewer's insightful critique, which has helped us refine our mechanistic interpretation. DSS treatment has been well-studied as a model for IBD, a 3-10% DSS solution was initially used to induce acute colitis in mice, resulting in characteristic phenotypes including diarrhea, bloody stools, and body weight loss (Okayasu *et al*, 1990). To study the function of VGLL4 in small intestine, first we used an irradiation model to treat mice. We also referred to a study in which 2% DSS was administered in drinking water for five consecutive days, followed by 3-day regular drinking water to induce ileitis (Cerantola *et al*, 2022). In Figure 2A, they evaluated the damage in ileum samples by H&E staining, as detailed below. The reduction in small intestine and colon length, along with inflammation was induced by DSS treatment. Another study by the same group reached similar conclusions (Faggin *et al*, 2025). Another study also used the combination of different diet and DSS treatment to induce ileitis in Figure 1B (Zheng *et al*, 2024), as detailed below. The impact of DSS on the small intestine has also been reported in several other studies (Geier *et al*, 2008; Lykov *et al*, 2018; Ohtsuka, 2003; Pai *et al*, 2014; Yazbeck *et al*, 2011).

Here panel 2A from Cerantola et al. (2022) was shown. Cerantola S, Faggin S, Caputi V, Bosi A, Banfi D, Rambaldo A, Porzionato A, Di Liddo R, De Caro R, Savarino EV et al (2022) Small intestine neuromuscular dysfunction in a mouse model of dextran sulfate sodium-induced ileitis: Involvement of dopaminergic neurotransmission. Life Sciences 301:120562.

Here panel 1B from Zheng et al. (2024) was shown. Zheng C, Wang L, Zou T, Lian S, Luo J, Lu Y, Hao H, Xu Y, Xiang Y, Zhang X et al (2024) Ileitis promotes MASLD progression via bile acid modulation and enhanced TGR5 signaling in ileal CD8 + T cells. Journal of Hepatology 80: 764-777.

We fully agree with your guess that the impact on the small intestines in this model likely reflects a secondary effect caused by the systemic illness of the mice. DSS treatment is a complex process and can be influenced by various factors, such as defensins and lysozyme, which are produced by Paneth cells. An elegant study proved that the deficiency in Paneth cell defensin production due to DEAH-box RNA helicase 15 (Dhx15) deletion showed more severe injury upon DSS treatment in colon. They used both Villin-Cre and Defa6-Cre and had the same results, indicating that the dysbiosis of the intestinal microbiota caused by α -defensins plays a significant role in regulating intestinal inflammation (Wang *et al*, 2021). Same conclusions were drawn in other studies, where DSS-induced colitis in mice was obviously alleviated by treatment with recombinant DEFA1, a major α -defensin secreted by Paneth cells (Hu *et al*, 2019). Furthermore, an engineered probiotic producing human Defensin-5 was shown to alleviate DSS-induced mucosal damage (Zeng *et al*, 2020), and systemically administered recombinant human β -defensin 2 was found to ameliorate DSS-induced intestinal inflammation and body weight loss (Koeninger *et al*, 2020).

Given potential variations in mouse age, housing conditions, and genetic background reported in previous studies, we optimized our sample collection timeline based on preliminary data, monitoring body weight loss and hematochezia severity in response to DSS treatment. We repeated the DSS-treatment assay and evaluate the impact of DSS on both small intestine and colon. The results showed that *Vgll4*^{IEC-KO} mice exhibited more severe intestinal injury, as reflected by greater body weight loss (Figure 2H). Although the overall length of the small intestine showed no significant change (Figure 2I-2J), morphological analysis revealed shorter villi length in the ileum (Figure 2K-2L), along with a reduction in colon length (Figure 2M-2N) and more severe edema and inflammatory infiltration (Figure 2O). These data indicate that DSS treatment also impaired the small intestine, manifested by disrupted villi structure, decreased villi height, and pronounced inflammatory infiltration. VGLL4 deficiency exacerbated DSS-induced injury in both the small intestine and colon, further supporting an essential role for VGLL4 in intestinal regeneration. Our study also indicated that VGLL4 loss impaired the antimicrobial peptide production in Paneth cells. Considering the relationship between antimicrobial peptides and intestinal inflammation, VGLL4 loss could regulate the DSS-induced colitis by regulating Paneth cells. Previously, we also found that in colonic cancer cells, VGLL4 regulates Wnt signaling targets by forming a complex with TEAD4 and TCF4. Thus, VGLL4 may have the potential to regulate colonic stem cell proliferation, and our collaborators are currently investigating VGLL4's function in colon inflammation. Overall, we believe that VGLL4 regulates DSS-induced colitis, partially by regulating Paneth cells and the intestinal microenvironment. We are grateful for your insightful comments, which have helped us refine our interpretations and improve the manuscript.

Figure 2

(Figure 2H) Body weight of *Vgll4*^{fl/fl} and *Vgll4*^{IEC-KO} mice during DSS treatment (n=6-8). (Figure 2I) Representative images of small intestines from *Vgll4*^{fl/fl} and *Vgll4*^{IEC-KO} mice after DSS treatment. Scale bar, 1 cm. (Figure 2J) Quantification of small intestine length after DSS (n=6-8). (Figure 2K) H&E staining of the small intestines from *Vgll4*^{fl/fl} and *Vgll4*^{IEC-KO} mice. Scale bars in left panel, 1 mm. Scale bars in right panel, 50 μ m. (Figure 2L) Quantification of small intestinal villi length after DSS (n=6-8). (Figure 2M) Representative images of colon from *Vgll4*^{fl/fl} and *Vgll4*^{IEC-KO} mice after DSS treatment. Scale bar, 1 cm. (Figure 2N) Quantification of colon length after DSS (n=6-8). (Figure 2O) H&E staining of the colon samples from *Vgll4*^{fl/fl} and *Vgll4*^{IEC-KO} mice. Scale bars, 200 μ m. Results are shown as mean + SD. *p < 0.05, **p < 0.01, ***p < 0.001, ****p < 0.0001. ns: no significance. Unpaired Student's t-test.

3. From Fig. 1, VGLL4 is expressed in both Paneth cells and non-Paneth crypt cells, including the stem cells. Therefore, the observed phenotypes in both injury models may be directly caused by a stem cell defect rather than Paneth cells. However, the stem cells in these mice are not well characterized.

We sincerely appreciate your constructive suggestions. To further investigate the role of VGLL4 in intestinal stem cells (ISCs), we performed two sets of experiments to detect the role of VGLL4 in stem cells under homeostasis stage and in vitro stage.

First, in our previous work, we got a mouse line *Vgll4^{fl/-};LSL-RFP* that included both *Vgll4*-minus and *Rosa-fsRfp* (Yu *et al*, 2019). Using this mouse model we generated a stem cell-specific *Vgll4* knockout line by crossing with *Lgr5-creERT2-eGFP^{+/-}* mice. The resulting ISC-KO mouse line (*Lgr5-creERT2-eGFP^{+/-}*, *Vgll4^{fl/-};LSL-RFP*) were compared to control mice (*Lgr5-creERT2-eGFP^{+/-}*, *Ai9^{+/-}*) (Response Figure 1A). Both groups received tamoxifen (TAM) injections for 7 days before small intestinal tissues were collected. IF staining confirmed knockout efficiency in the crypts (Response Figure 1B), where *Lgr5*-creERT2 activity-driven RFP expression was accompanied by a loss of VGLL4 signal in ISC-KO crypts, indicating successful colonization by VGLL4-negative stem cells. We next examined whether VGLL4 deletion in ISCs affected cell proliferation. The number of Ki67⁺ cells in RFP-positive crypts was comparable between control and ISC-KO mice (Response Figure 1C-1D), suggesting that VGLL4 loss does not impair the proliferative capacity of ISCs *in vivo*. In contrast, the proportion of Paneth cells was significantly reduced in VGLL4-deficient crypts (Response Figure 1E-1F), supporting the hypothesis that VGLL4 is required for Paneth cell maintenance. Together, these results indicate that conditional knockout of VGLL4 in intestinal stem cells does not affect their proliferation under homeostatic conditions but leads to a decrease in Paneth cell numbers.

Response Figure 1

A

Control: *Lgr5-creERT2-eGFP*, *Ai9^{+/-}*

TAM Injection
Day 7

SI samples

ISC-KO: *Lgr5-creERT2-eGFP*, *Vgll4^{fl/-};LSL-RFP*

B

(Response Figure 1A) Model of Control (*Lgr5-creERT2-eGFP^{+/-}*, *Ai9^{+/-}*) mouse line and ISC-KO (*Lgr5-creERT2-eGFP^{+/-}*, *Vgll4^{fl/-}-LSL-RFP*) mouse line construction strategies and small intestinal samples harvest strategy. (Response Figure 1B) IF of VGLL4 (gray), GFP (green), tdTomato (purple) and DAPI (blue) in Control and ISC-KO mouse small intestinal samples after TAM injection for a week. Scale bars, 30 μ m. (Response Figure 1C) IF of Ki67 (gray), GFP (green), tdTomato (purple) and DAPI (blue) in Control and ISC-KO mouse small intestinal samples after TAM injection for a week. Scale bars, 10 μ m. (Response Figure 1D) Quantification of Ki67⁺ cells per crypts in (Response Figure 1C) (n=4). (Response Figure 1E) IF of Lysozyme (gray), GFP (green), tdTomato (purple) and DAPI (blue) in Control and ISC-KO mouse small intestinal samples after TAM injection for a week. Scale bars, 10 μ m. (Response Figure 1F) Quantification of Lysozyme⁺ cells per crypts in (Response Figure 1E) (n=4).

Next, to further study the role of VGLL4 in stem cell in vitro, we performed the co-culture assay. Wild-type *Lgr5⁺* stem cells (WT *Lgr5⁺*) were sorted from *Lgr5-creERT2-eGFP^{+/-}* mice, and VGLL4 knockout *Lgr5⁺* stem cells (VGLL4-KO *Lgr5⁺*) were sorted from *Lgr5-creERT2-eGFP^{+/-}*, *Vgll4^{fl/fl}* mice. All *Lgr5-creERT2-eGFP^{+/-}* mice and *Lgr5-creERT2-eGFP^{+/-}*, *Vgll4^{fl/fl}* mice were injected with TAM three times within one week. Wild-type Paneth cells (WT PCs) were sorted from *Vgll4^{fl/fl}* mice, and VGLL4 knockout Paneth cells (VGLL4-KO PCs) were sorted from *Vgll4^{IEC-KO}* mice. We set four groups: WT *Lgr5⁺* with WT PCs, WT *Lgr5⁺* with VGLL4-KO PCs, VGLL4-KO *Lgr5⁺* with WT PCs, VGLL4-KO

Lgr5⁺ with VGLL4-KO PCs. WT Lgr5⁺ cells co-cultured with WT PCs resulted in the largest number and diameter of organoids, while VGLL4-KO Lgr5⁺ cells co-cultured with VGLL4-KO PCs formed the fewest and smallest organoids. Compared to the organoids formed by WT Lgr5⁺ cells co-cultured with WT PCs, VGLL4-KO Lgr5⁺ cells co-cultured with WT PCs showed a slight decrease in organoid number, and exhibited a tendency towards smaller diameters (Response Figure 2A-C).

These findings suggest that VGLL4 may functionally regulate stem cells. However, VGLL4 knockout in intestinal crypts did not affect proliferation in vivo (Response Figure 1C-D, Figure EV4G-H). Co-culture of WT Lgr5⁺ cells with VGLL4-KO PCs formed smaller organoids than those formed by VGLL4-KO Lgr5⁺ cells with WT PCs, indicating that the knockout of VGLL4 in PCs had a greater influence on organoid formation than in Lgr5⁺ stem cells, prompting us to study VGLL4's role in Paneth cells.

Response Figure 2

(Response Figure 2A) Representative pictures of organoids from wild type Lgr5⁺ stem cells (WT Lgr5⁺) or VGLL4 Knockout Lgr5⁺ stem cells (VGLL4-KO Lgr5⁺) co-cultured with Paneth cells from *Vgll4^{fl/fl}* and *Vgll4^{IEC-KO}* mice for 5 days. Scale bars, 60 μ m. (Response Figure 2B) Quantification of organoids number in (Response Figure 3A) (n=6). (Response Figure 2C) Quantification of organoids diameter in (Response Figure 2A) (n=6). Results are shown as mean + SD. *p < 0.05, **p < 0.01, ***p < 0.001, ****p < 0.0001. ns: no significance. B-C, One-way Anova.

Our findings suggest that the niche microenvironment may functionally compensate for VGLL4 deficiency in maintaining intestinal stem cell proliferation in vivo, whereas this compensatory capacity is lost under in vitro conditions, resulting in measurable proliferation changes following VGLL4 depletion.

4. The biochemistry in Fig. 4 was well done. However, the significance of these results is not very obvious.

We appreciate your insightful comments. In Fig.4, we elucidated the formation mechanism of the VGLL4-TEAD4-ATOH1 complex and identified the key active sites involved in its assembly. We also confirmed that the interaction was independent of DNA by adding DNase to the co-IP system and by using purified proteins. The consistency of these findings—demonstrated through multiple independent experiments (Figure 4E-I, Figure EV5C-H, and Response Figure 3)—provides strong and reproducible evidence for the existence and molecular basis of the VGLL4-TEAD4-ATOH1 complex. The discovery of the VGLL4-TEAD4-ATOH1 transcriptional complex represents a significant advance by elucidating a novel mechanism that fundamentally expands the functional scope of the ATOH1 transcription factor. Our findings establish that the assembly of this ternary complex is critical for initiating the terminal differentiation program in Paneth cells. This provides a direct molecular explanation for a key step in intestinal epithelial development.

(Figure 4E) Co-immunoprecipitation assay results between ATOH1 and TEAD4 in HEK293T cells. (Figure 4F) Co-immunoprecipitation assay results between ATOH1 and TEAD4 in HEK293T cells with DNase. (Figure 4G) Co-immunoprecipitation assay results between ATOH1 and different mutations of TEAD4 in HEK293T cells. (Figure 4H) Co-immunoprecipitation assay results between ATOH1 and VGLL4 in HEK293T cells with or without TEAD4. (Figure 4I) Immunoblot analysis of GST pull-down assay between purified proteins ATOH1-ΔN, VGLL4, and TEAD4.

Response Figure 3

(Response Figure 3A) Co-immunoprecipitation assay results between ATOH1 and TEAD4 in HEK293T cells. (Response Figure 3B) Co-immunoprecipitation assay results between ATOH1 and VGLL4 in HEK293T cells with or without TEAD4. (Response Figure 3C) Co-immunoprecipitation assay results between ATOH1 and TEAD4 wildtype or 4M mutant form in HEK293T cells.

The functional implications of our work extend beyond the intestinal epithelium, offering broader insights into the biological roles of the proteins involved. ATOH1 and its downstream effector GF11 are well-established as essential transcription factors for hair cell development in the cochlea (Iyer et al., 2022; McGovern et al., 2024). Similarly, the YAP/TEAD complex has been demonstrated to regulate the self-renewal gene network in progenitor cells of the organ of Corti (Gnedeva et al., 2020). While the role of VGLL4 in the inner ear remains uncharacterized, our finding that the VGLL4–TEAD4–ATOH1 complex regulates GF11 expression strongly suggests that VGLL4 likely plays a critical and previously overlooked role in cochlear development.

More broadly, our study redefines the transcriptional function of the Hippo pathway effector TEAD4. We demonstrate that the VGLL4–TEAD4 interaction serves as a molecular platform for recruiting the pro-differentiation factor ATOH1, thereby redirecting TEAD4 from a promoter of proliferation to an initiator of cellular maturation. This represents a paradigm shift in understanding TEAD4 function, revealing that its transcriptional output is not fixed but is dynamically determined by the specific co-factors it recruits.

The discovery of the complex also opens new biological and therapeutic perspectives. It raises the possibility that the pathway we identified may direct cell fate decisions in contexts beyond the intestine. A recent study showed that colorectal cancer cells can undergo transdifferentiation into a Paneth cell-like phenotype to acquire drug resistance (Zhang et al., 2025). This observation prompts the question of what role of the VGLL4-TEAD4-ATOH1 complex plays in this process—specifically, whether its activation participates in or modulates this transdifferentiation pathway—which remains unknown and represents a key direction for our future research.

In summary, our study not only identifies a novel transcriptional regulatory complex essential for Paneth cell biology, but also establishes a broader paradigm for VGLL4-mediated transcriptional regulation with implications for epithelial homeostasis across multiple organs. These findings provide a transformative perspective for future research, shifting the focus from single developmental pathways toward therapeutically reprogramming transcriptional complexes to restore healthy cell identity.

References:

Cerantola S, Faggin S, Caputi V, Bosi A, Banfi D, Rambaldo A, Porzionato A, Di Liddo R, De Caro R, Savarino EV *et al* (2022) Small intestine neuromuscular dysfunction in a mouse model of dextran sulfate sodium-induced ileitis: Involvement of dopaminergic neurotransmission. *Life Sciences* 301

Faggin S, Cerantola S, Caputi V, Tietto A, Stocco E, Bosi A, Ponti A, Bertazzo A, Macchi V, Porzionato A *et al* (2025) Toll-like receptor 4 deficiency ameliorates experimental ileitis and enteric neuropathy: Involvement of nitrenergic and 5-hydroxytryptaminergic neurotransmission. *British Journal of Pharmacology* 182: 1803-1822

Geier MS, Smith CL, Butler RN, Howarth GS (2008) Small-Intestinal Manifestations of Dextran Sulfate Sodium Consumption in Rats and Assessment of the Effects of Lactobacillus fermentum BR11. *Digestive Diseases and Sciences* 54: 1222-1228

Gnedeva K, Wang X, McGovern MM, Barton M, Tao L, Trecek T, Monroe TO, Llamas J, Makmura W, Martin JF *et al* (2020) Organ of Corti size is governed by Yap/Tead-mediated progenitor self-renewal. *Proceedings of the National Academy of Sciences* 117: 13552-13561

Hu X, Deng J, Yu T, Chen S, Ge Y, Zhou Z, Guo Y, Ying H, Zhai Q, Chen Y *et al* (2019) ATF4 Deficiency Promotes Intestinal Inflammation in Mice by Reducing Uptake of Glutamine and Expression of Antimicrobial Peptides. *Gastroenterology* 156: 1098-1111

Iyer AA, Hosamani I, Nguyen JD, Cai T, Singh S, McGovern MM, Beyer L, Zhang H, Jen HI, Yousaf R *et al* (2022) Cellular reprogramming with ATOH1, GFI1, and POU4F3 implicate epigenetic changes and cell-cell signaling as obstacles to hair cell regeneration in mature mammals. *Elife* 11

Koeninger L, Armbruster NS, Brinch KS, Kjaerulf S, Andersen B, Langnau C, Autenrieth SE, Schneidawind D, Stange EF, Malek NP *et al* (2020) Human β -Defensin 2 Mediated Immune Modulation as Treatment for Experimental Colitis. *Frontiers in Immunology* 11

Lykov AP, Poveshchenko OV, Bondarenko NA, Surovtseva MA, Kim II, Bgatova NP (2018) Therapeutic Potential of Biomedical Cell Product in DSS-Induced Inflammation in the Small Intestine of C57Bl/6J Mice. *Bulletin of Experimental Biology and Medicine* 165: 576-580

McGovern MM, Hosamani IV, Niu Y, Nguyen KY, Zong C, Groves AK (2024) Expression of Atoh1, Gfi1, and Pou4f3 in the mature cochlea reprograms nonsensory cells into hair cells. *Proc Natl Acad Sci U S A* 121: e2304680121

Ohtsuka Y (2003) Dextran Sulfate Sodium--Induced Inflammation Is Enhanced by Intestinal Epithelial Cell Chemokine Expression in Mice. *Pediatric Research* 53: 143-147

Okayasu I, Hatakeyama S, Yamada M, Ohkusa T, Inagaki Y, Nakaya R (1990) A novel method in the induction of reliable experimental acute and chronic ulcerative colitis in mice. *Gastroenterology* 98: 694-702

Pai MH, Liu JJ, Hou YC, Yeh CL (2014) Soybean and Fish Oil Mixture With Different ω -6/ ω -3 Polyunsaturated Fatty Acid Ratios Modulates Dextran Sulfate Sodium--Induced Changes in Small Intestinal Intraepithelial $\gamma\delta$ T-Lymphocyte Expression in Mice. *Journal of Parenteral and Enteral Nutrition* 40: 383-391

Wang Y, He K, Sheng B, Lei X, Tao W, Zhu X, Wei Z, Fu R, Wang A, Bai S *et al* (2021) The RNA helicase Dhx15 mediates Wnt-induced antimicrobial protein expression in Paneth cells. *Proceedings of the National Academy of Sciences* 118

Yazbeck R, Howarth GS, Butler RN, Geier MS, Abbott CA (2011) Biochemical and histological changes in the small intestine of mice with dextran sulfate sodium colitis. *Journal of Cellular Physiology* 226: 3219-3224

Yu W, Ma X, Xu J, Heumuller AW, Fei Z, Feng X, Wang X, Liu K, Li J, Cui G *et al* (2019) VGLL4 plays a critical role in heart valve development and homeostasis. *PLoS Genet* 15: e1007977

Zeng L, Tan J, Xue M, Liu L, Wang M, Liang L, Deng J, Chen W, Chen Y (2020) An engineering probiotic producing defensin-5 ameliorating dextran sodium sulfate-induced mice colitis via Inhibiting NF-kB pathway. *Journal of Translational Medicine* 18

Zhang Y, Chen J, She Y, Fang Z, Zhang Y, Ruan D, Guo W, Liao J, Zhou W, Lao J *et al* (2025) Paneth-like transition drives resistance to dual targeting of KRAS and EGFR in colorectal cancer. *Cancer Cell*

Zheng C, Wang L, Zou T, Lian S, Luo J, Lu Y, Hao H, Xu Y, Xiang Y, Zhang X *et al* (2024) Ileitis promotes MASLD progression via bile acid modulation and enhanced TGR5 signaling in ileal CD8+ T cells. *Journal of Hepatology* 80: 764-777

Dear Dr. Zhang,

Thank you for the submission of your further revised manuscript to our editorial offices. I have now received the report from the referee that I asked to re-evaluate the study, you will find below. As you will see, the referee now supports publication of your study in EMBO reports. However, the referee has three remaining concerns and suggestions to improve the study, I ask you to address in a final revised manuscript. Importantly, as indicated in point 3, the data presented in the previous p-b-p-response needs to be incorporated into the manuscript. Please also update the source data of any figure that is added to the submission.

Please also provide a final p-b-p-response regarding the remaining referee points and the editorial requests below.

Editorial requests:

- Please note that all corresponding authors are required to supply an ORCID ID for their name upon submission of a revised manuscript. We will not proceed with publication if this is not done. The ORCID of authors Lei Zhang is still missing. The ORCID ID needs to be linked to the author account in our manuscript tracking system. This can only be done by the author.

- Please order the manuscript sections like this, using only these names:

Title page - Abstract - Keywords - Introduction - Results - Discussion - Methods - Data availability section - Acknowledgements (please include here all the funding information) - Disclosure and Competing Interests Statement - References - Figure legends - Expanded View Figure legends

- Please check again that the number "n" for how many independent experiments were performed, their nature (biological versus technical replicates), the bars and error bars (e.g. SEM, SD) and the test used to calculate p-values is indicated in the respective figure legends (main and EV figures). Please also check that all the p-values are explained in the legend, and that these fit to those shown in the figure. Please provide statistical testing where applicable. Please avoid the phrase 'independent experiment' but clearly state if these were biological or technical replicates. Please also indicate (e.g. with n.s.) if testing was performed, but the differences are not significant. In case n=2, please show the data as separate datapoints without error bars and statistics. See also:

<https://link.springer.com/journal/44319/submission-guidelines#cms-Figure-and-data-presentation>

If n<5, please show single datapoints for diagrams. Moreover:

- Please note that the exact p values are not provided in the legends of figures 1d,h; 2a,d,g,h,l,n; 3c,e,g,j,k; 4a,c; 5d-k; 6b,d,e,g,h,i,j,l-p; EV-1c,d; EV-4j-l,n,o; EV-5b-h.

- Please indicate the statistical test used for data analysis in the legends of figures 1b; 3a.

- Please note that the box plots need to be defined in terms of minima, maxima, centre, bounds of box and whiskers, and percentile in the legends of figures EV-1d; EV-6e,f.

- Although 'n' is provided, please describe the nature of entity for 'n' in the legends of figures 1d,h.

- Please note that the measure of center for the error bars needs to be defined in the legends of figures EV-6l.

- Please note that the red squares are not defined in the legend of figure EV-1b. This needs to be rectified.

- Please note that the dotted lines are not defined in the legend of figure 6a. This needs to be rectified.

- Please add all primer information (presently the tables S1-S3 in the Appendix) to the Reagents & Tools Table and remove the table from the main manuscript text. The R& T table must be uploaded only separately. Please add call outs to the Reagents & Tools table where appropriate. Finally, please remove the Appendix file.

- Please confirm that for all Western blot panels (main, EV, or Appendix figures) the loading control was run on the same gel as the other proteins detected. Please note that we discourage comparisons between samples on different gels/blots, even if the samples derive from one experiment, as confounding factors reduce comparability. If unavoidable, the figure legend must state that the samples derive from the same experiment and that gels/blots were processed in parallel. If a 'representative' loading control is shown for multiple gels/blots, the intra-gel controls should be shown in the source data files, and the figure legends should describe the data displayed accurately. See our author guidelines:

<https://link.springer.com/journal/44319/submission-guidelines#cms-Figure-and-data-presentation> (section 'Electrophoretic gels and blots').

I look forward to seeing the further revised version of your manuscript when it is ready. Please let me know if you have questions regarding the revision.

Best,

Referee #2:

I thank the authors for their responses to my prior critiques. Here are my major comments:

1. In response to my first comment, the authors revised their narrative, but still claimed in the Abstract: "Consistently, a remarkable reduction in VGLL4 expression is observed in human ulcerative colitis specimens." I suggest that the authors tone down their statement. First, no human ulcerative colitis specimen was analyzed in their study, the ulcerative colitis specimens here refer to the public database. Second, for the acute inflammation patient samples that were indeed analyzed by the authors, the N numbers were quite low (5 vs. 9) and shall not be considered as "a remarkable reduction".

2. My second point was "The manuscript tried to link the disease susceptibility of Vgll4 intestinal knockout mice to Paneth cell defect, as suggested by the title. But the experiments could not establish a causality."
The authors provided lengthy responses to elaborate on the importance of Paneth cells and Paneth cell derived antimicrobial peptides, such as defensin, in intestinal inflammation. However, these cited studies supported the importance of Paneth cells, but they did not establish the necessity of Paneth cells in the current context of Vgll4 and gut homeostasis or inflammation, which are studied here by the authors.

For example, the authors only cited work that showed a positive protection by Paneth cell-derived defensin against inflammation. The authors did not cite or were not aware of protections that can also result from lacking of some of the Paneth cell derived factors, such as lysozyme, Reg proteins, etc.

Based on the title, readers expect that Vgll4 regulates gut homeostasis through Paneth cells. However, the existing data showed that Vgll4 regulates gut homeostasis and Paneth cell numbers. A causality could be established if rescue experiments involving Paneth cells or their production of defensin in the knockout mice are performed. Otherwise, the title may be revised to accurately reflect the data.

3. In response to my third point "VGLL4 is expressed in both Paneth cells and non-Paneth crypt cells, including the stem cells. Therefore, the observed phenotypes in both injury models may be directly caused by a stem cell defect rather than Paneth cells. However, the stem cells in these mice are not well characterized."

The authors included some new data labeled as "Response Figures". In these response figures, the authors showed some analysis of stem cell specific VGLL4 knockout mice. They also performed coculture experiments using different combinations of WT or KO Paneth cells + WT or KO stem cells. These data (Response figures) should be carefully presented, quantified, and included in the main figures.

Referee #2:

I thank the authors for their responses to my prior critiques. Here are my major comments:

1. In response to my first comment, the authors revised their narrative, but still claimed in the Abstract: "Consistently, a remarkable reduction in VGLL4 expression is observed in human ulcerative colitis specimens." I suggest that the authors tone down their statement. First, no human ulcerative colitis specimen was analyzed in their study, the ulcerative colitis specimens here refer to the public database. Second, for the acute inflammation patient samples that were indeed analyzed by the authors, the N numbers were quite low (5 vs. 9) and shall not be considered as "a remarkable reduction".

Thank you for your rigorous and constructive comments, which have helped us further refine the accuracy and rigor of our manuscript.

We fully agree with your point regarding the precision of the statement in the Abstract. Following your suggestion, we have revised the original sentence to: "**Consistently, a decrease in VGLL4 expression is observed in public databases of human colitis samples.**" This revision explicitly clarifies that the human colitis sample data were derived from public databases (rather than samples analyzed de novo in our study) and adjusts the wording to avoid overstating the observation—addressing both concerns you raised about the source of specimens and the appropriateness of descriptive language given the sample size limitations of our own acute inflammation analyses.

2. My second point was "The manuscript tried to link the disease susceptibility of Vgll4 intestinal knockout mice to Paneth cell defect, as suggested by the title. But the experiments could not establish a causality."

The authors provided lengthy responses to elaborate on the importance of Paneth cells and Paneth cell derived antimicrobial peptides, such as defensin, in intestinal inflammation. However, these cited studies supported the importance of Paneth cells, but they did not establish the necessity of Paneth cells in the current context of Vgll4 and gut homeostasis or inflammation, which are studied here by the authors.

For example, the authors only cited work that showed a positive protection by Paneth cell-derived defensin against inflammation. The authors did not cite or were not aware of protections that can also result from lacking of some of the Paneth cell derived factors, such as lysozyme, Reg proteins, etc.

Based on the title, readers expect that Vgll4 regulates gut homeostasis through Paneth cells. However, the existing data showed that Vgll4 regulates gut homeostasis and Paneth cell numbers. A causality could be established if rescue experiments involving Paneth cells or their production of defensin in the knockout mice are performed. Otherwise, the title may be revised to accurately reflect the data.

Thank you for your critical and insightful comment regarding the causal link between VGLL4 deletion, Paneth cell defects, and intestinal homeostasis, as well as the alignment of our title with the current data. We acknowledge that the current study only establishes a correlation between *Vgll4* depletion, reduced Paneth cell proportions, and impaired gut homeostasis, rather than a definitive causal relationship. As suggested, direct evidence supporting Paneth cells as the key mediator of *Vgll4*-regulated intestinal function would

require targeted rescue experiments—such as restoring Paneth cell function or replenishing Paneth cell-derived factors in *Vgll4* intestinal knockout mice. We agree that this represents a major limitation of our research. Future investigations will focus on these rescue strategies to clarify whether Paneth cell defects are both necessary and sufficient for the gut homeostasis phenotypes observed in *Vgll4*-deficient mice. To accurately reflect the correlative nature of our findings, we have revised the title to remove any implication of a direct causal mechanism: “**VGLL4 Modulates Paneth Cells and Sustains Intestinal Homeostasis**”. This revised title retains our core findings that VGLL4 affects intestinal homeostasis and Paneth cell abundance, while avoiding overstatements about the relationship that has not yet been definitively proven. We have carefully revised sections of the Results and Discussion where we previously implied a direct causal link to strictly reflect the correlative nature of our current data, ensuring scientific rigor in our conclusions.

3. In response to my third point "VGLL4 is expressed in both Paneth cells and non-Paneth crypt cells, including the stem cells. Therefore, the observed phenotypes in both injury models may be directly caused by a stem cell defect rather than Paneth cells. However, the stem cells in these mice are not well characterized."

The authors included some new data labeled as "Response Figures". In these response figures, the authors showed some analysis of stem cell specific VGLL4 knockout mice. They also performed coculture experiments using different combinations of WT or KO Paneth cells + WT or KO stem cells. These data (Response figures) should be carefully presented, quantified, and included in the main figures.

Thank you for your constructive comment regarding the presentation of our stem cell-related data. We fully agree with your suggestion that the newly generated data should be carefully organized and integrated into the main figures to improve the clarity and narrative coherence.

In response, we have revised the manuscript's figure layout as follows:

We have incorporated the *Vgll4* stem cell-specific KO mouse analysis into **Figures EV3G-EV3H**, **Figures 4A-4B**, and logically reorganized the panel structure. The co-culture experiment data have been added to **Figures 3I-3K** with clear subpanel labels to distinguish different experimental combinations. All detailed quantification in these datasets were supplemented. Also, we have updated the legends of Figures EV2-EV3 and Figure 3-4 to explicitly describe the experimental design and quantification methods, to ensure clarity and reproducibility.

These changes not only directly address your concern regarding data presentation but also further clarify the function of VGLL4 in intestinal stem cells, strengthening the mechanistic depth of our study and substantially enhancing the overall rigor and completeness of the manuscript. We hope these revisions align with your expectations.

Dr. Lei Zhang
Shanghai Jiao Tong University
Sheng Yushou Center of Cell Biology and Immunology
China

Dear Dr. Zhang,

Thank you for the submission of your final revised manuscript to our editorial offices. It now went through this and your final p-b-p-response and consider the remaining points of referee #2 and the editorial requests as adequately addressed.

I am thus very pleased to accept your manuscript for publication in the next available issue of EMBO reports. Thank you for your contribution to our journal.

You may qualify for financial assistance for your publication charges - either via a Springer Nature fully open access agreement or an EMBO initiative. Check your eligibility: <https://link.springer.com/journal/44319/how-to-publish-with-us>

Yours sincerely,

>>> Please note that it is EMBO Reports policy for the transcript of the editorial process (containing referee reports and your response letter) to be published as an online supplement to each paper. If you do NOT want this, you will need to inform the Editorial Office via email immediately. More information is available here: <https://link.springer.com/partners/embo-press/editorial-policies#Peer%20review>